# Prediction of perinatal death using machine learning models: a birth registry-based cohort study in northern Tanzania

Innocent B Mboya [1,2] Michael J Mahande [2] Mohanad Mohammed,[1] Joseph Obure,[3] Henry G Mwambi[1]

► Prepublication history and supplemental material for this paper are available online. To view these files, please visit the journal online (http://dx.doi.org/10.1136/bmjopen-2020-040132).

[1]School of Mathematics, Statistics, and Computer Science, University of KwaZulu-Natal, Pietermaritzburg, KwaZulu-Natal, South Africa
[2]Department of Epidemiology and Biostatistics, Institute of Public Health, Kilimanjaro Christian Medical University College, Moshi, Tanzania
[3]Department of Obstetrics and Gynecology, Kilimanjaro Christian Medical Center, Moshi, Tanzania

**Correspondence to**
Innocent B Mboya;
ib.mboya@gmail.com

## ABSTRACT

**Objective** We aimed to determine the key predictors of perinatal deaths using machine learning models compared with the logistic regression model.

**Design** A secondary data analysis using the Kilimanjaro Christian Medical Centre (KCMC) Medical Birth Registry cohort from 2000 to 2015. We assessed the discriminative ability of models using the area under the receiver operating characteristics curve (AUC) and the net benefit using decision curve analysis.

**Setting** The KCMC is a zonal referral hospital located in Moshi Municipality, Kilimanjaro region, Northern Tanzania. The Medical Birth Registry is within the hospital grounds at the Reproductive and Child Health Centre.

**Participants** Singleton deliveries (n=42 319) with complete records from 2000 to 2015.

**Primary outcome measures** Perinatal death (composite of stillbirths and early neonatal deaths). These outcomes were only captured before mothers were discharged from the hospital.

**Results** The proportion of perinatal deaths was 3.7%. There were no statistically significant differences in the predictive performance of four machine learning models except for bagging, which had a significantly lower performance (AUC 0.76, 95% CI 0.74 to 0.79, p=0.006) compared with the logistic regression model (AUC 0.78, 95% CI 0.76 to 0.81). However, in the decision curve analysis, the machine learning models had a higher net benefit (ie, the correct classification of perinatal deaths considering a trade-off between false-negatives and false-positives)—over the logistic regression model across a range of threshold probability values.

**Conclusions** In this cohort, there was no significant difference in the prediction of perinatal deaths between machine learning and logistic regression models, except for bagging. The machine learning models had a higher net benefit, as its predictive ability of perinatal death was considerably superior over the logistic regression model. The machine learning models, as demonstrated by our study, can be used to improve the prediction of perinatal deaths and triage for women at risk.

## Strengths and limitations of this study

► The Kilimanjaro Christian Medical Centre (KCMC) Medical Birth Registry cohort data provide a rich source of information for monitoring trends, inform clinical and administrative decisions, and enables complex modelling of the key predictors of perinatal deaths, among other adverse pregnancy outcomes.

► While standard regression models such as logistic regression are extensively applied in the literature to predict adverse pregnancy outcomes such as perinatal deaths, the application of machine learning models is limited.

► Machine learning algorithms may improve the prediction ability of perinatal deaths, and enable triage of women at high risk of experiencing adverse perinatal outcomes.

► The birth registry only captured deaths occurring in the KCMC hospital hence might have underestimated the proportion of perinatal deaths.

## INTRODUCTION

Neonatal survival is at the heart of Sustainable Development Goals agenda.[1 2] The Every Newborn Action Plan to end Preventable Deaths set a goal for all countries to reach the target of ten or less newborn deaths per 1000 live births and 10 or less stillbirths per 1000 total births by the year 2035.[3] Furthermore, the United Nations set the target of reducing neonatal mortality to 12 deaths per 1000 live births or fewer by 2030.[1] Globally, neonatal deaths declined by 51% from 5 million in 1990 to 2.5 million in 2017. But this decline has not been realised in low-income and middle-income countries, which carries the highest burden of neonatal deaths, with south Asia and sub-Saharan Africa accounting for 79% of the total burden of neonatal deaths in 2017.[4] Furthermore, the under-5 mortality rate has decreased almost across the world, but the proportions of neonatal deaths remained high in this group.[5 6] Neonatal deaths accounted for 47% of all under-5 deaths in 2018, and it has increased from 40% in 1990, with sub-Saharan Africa bearing the highest

burden.[6] Globally 2.5 million children died in the first month of life in 2018, with approximately 7000 newborn deaths every day.[6] Nearly three-quarters of these deaths occur during the first week, with about one million dying on the first day and close to one million dying within the next 6 days.[6]

Globally, more than five million perinatal deaths occur each year.[2] The majority (95%) of these deaths occur in sub-Saharan Africa and Southern Asia.[7] According to the Tanzania Demographic and Health Survey, the perinatal mortality rate has slightly increased from 36 to 39 deaths per 1000 live births between 2010–11 and 2015–16 survey rounds, respectively, relative to under-5 mortality.[8] In addition, perinatal mortality rate in Tanzania is the highest in East Africa.[7]

Early identification of pregnant women at risk for adverse maternal and perinatal outcomes during the prenatal period and timely provision of high-quality healthcare services have been reported to improve maternal and newborn survival.[9] Machine learning (hereafter denoted as 'ML') models are methodologies for developing algorithms that learn from existing data to make predictions on new data.[9] ML models have shown better predictive performance over the classical or conventional regression models,[10] and they can better handle a significant number of potential predictors. However, there is conflicting evidence of the performance of these models. Previous investigators have demonstrated that, compared with the classical regression models, ML models have superior performance for early differentiation of sepsis and non-infectious systemic inflammatory response syndrome in critically ill children,[11] in predicting neonatal and under-5 mortality,[12–16] and critical care and hospitalisation outcomes.[10 17 18] In contrast, other studies have shown no predictive performance benefit of the ML models in prediction of clinical outcomes.[9 19]

The first step in addressing high perinatal mortality is the accurate capture and classification of the causes of those deaths across all settings.[20] WHO International Classification of Diseases (ICD-10) is a standardised tool used for the classification of deaths occurring during the perinatal period: ICD-PM.[2 20 21] ML models may be an essential tool in the assessment of risk factors for deaths during the perinatal period and triage pregnant women at high risk of experiencing adverse perinatal outcomes, especially in low-resourced settings where the majority of perinatal deaths occur at home.[22–25] Capturing the chain of events that led to the perinatal mortality, from both the maternal and the perinatal side, informs the design and development of preventative and therapeutic measures.[2]

Using data from the medical birth registry at Kilimanjaro Christian Medical Centre (KCMC) referral hospital in northern Tanzania, we aimed to determine the key predictors of perinatal death using ML models. Previous studies using the same data[26–31] applied standard regression models to assess risk factors for adverse perinatal outcomes. A major weakness of conventional regression analysis, as opposed to ML models, is that many covariates

are excluded based on specific model assumptions. In contrast, ML techniques which are non-parametric in nature find the most predictive groupings of factors based on their frequency and strength of association, with no particular model assumptions.[32] In this study, we compared the predictive performance of the ML models with the conventional regression analysis, particularly logistic regression (Lreg).

## METHODS
### Study design, setting and population
We conducted a secondary analysis of birth cohort data from the KCMC referral hospital, situated in the Moshi Municipality of Kilimanjaro region, Northern Tanzania. The hospital receives deliveries from nearby communities and referral cases from other healthcare facilities inside the region and the neighbouring regions.[33] The hospital has an average annual delivery rate of 4000 births.[31 33 34] The study population was women who delivered singleton babies. We included 42 319 deliveries with complete records between 2000 and 2015. We excluded records with missing values on the outcome (perinatal status) and the covariates as well as pregnancies with multiple gestations to avoid over-representation of high-risk pregnancies.[31]

### Data source
We used data from the KCMC referral hospital medical birth registry between the years 2000 and 2015, which were collected among mothers who delivered at the department of obstetrics and gynaecology. More description of the KCMC medical birth registry is also available elsewhere.[26 28 31 35 36] Briefly, the KCMC medical birth registry is within hospital grounds at the Reproductive and Child Health Centre. The birth registry has been in operation since the year 2000, established to serve both clinical, administrative and research purposes.[35 36] Trained midwives collected data using a standardised questionnaire (within 24 hours after delivery or later in case a mother had recovered from complications), after which data are entered into a computerised database located at the birth registry. Also, additional data were abstracted from the antenatal care (ANC) cards and the hospital medical records of the mother.[28]

A unique hospital identification number was assigned to each woman at first admission and used to trace her medical records at later admissions, and further to link records of successive births of the same woman.[36] Data captured information on the background characteristics of mother and father, mother's health before and during present pregnancy, information about delivery including complications, and child characteristics including their status (ie, whether dead or alive).

### Study variables
The main outcome variable in this study was perinatal death which was defined as the number of stillbirths

(pregnancy loss that occurs after 7 months of gestation) and early neonatal deaths (deaths of live births within the first 7 days of life).[8 37] The perinatal death was coded as binary, that is, 'yes' if death occurred during the perinatal period and 'no' if otherwise. This outcome only captured deaths that occurred within the hospital before the discharge of mothers. There are no follow-up mechanisms for deaths that occur outside the health facility (KCMC hospital).

We included a total of 32 predictor variables for the ML models. Previous literature informed the selection of these variables,[4 27 38–45] most of which are available in the birth registry. These included maternal and paternal background characteristics; age in years, area of residence (rural vs urban), highest education level (none, primary, secondary and higher), marital status (single, married and widow/divorced) and occupation (unemployed, employed and others). Further, specific characteristics of the mother included referral status (whether referred for delivery or not), and the number of ANC visits (<4 and ≥4 visits).

We excluded maternal body mass index (BMI) and HIV status because they contributed to nearly 47% of all missing values in the dataset. Maternal health during pregnancy included; alcohol consumption, smoking, gestational diabetes, diabetes, hypertension, pre-eclampsia/eclampsia, bleeding (ie, the woman observed blood from the vagina at any time during the pregnancy), anaemia, malaria and systemic infections/sepsis. Variables with information concerning delivery included; induction of labour (yes or no), mode of delivery (vaginal vs caesarean section), presentation (breech vs cephalic), complications during birth, particularly premature rupture of the membranes, postpartum haemorrhage, placenta previa and placenta abruption, all categorised as yes and no. Gestational age at birth was estimated based on the date of the last menstrual period and recorded in full weeks. Preterm birth was defined as babies born alive before 37 weeks of pregnancy are completed.[46] Child characteristics included sex (male or female), low birthweight defined as an infant birth weight of less than 2500 g[27 47] and year of birth.

### Statistical and computational analysis

Data were cleaned and then analysed using Stata V.15.1.[48] Categorical variables were summarised using frequencies and proportions. The $\chi^2$ test statistic was used to test the relationships between a set of independent variables and perinatal death. For the ML models (ie, from feature selection, training, testing and comparison of the predictive performance of the machines), we used R V.3.6.3.[49] The training dataset contained 70% of randomly selected samples used to develop six different ML models to predict perinatal death. These are artificial neural networks (ANN), random forests (RF), Naïve Bayes (NB), bagged trees, boosting and the Lreg model. We used the *caret* package to implement these models in R.

Briefly, ANN is a method constructed from three layers of connected nodes: input, hidden and output.[50] The input where each input variable appears as a node; the hidden layer contains several nodes determined during the model tuning phase. In contrast, the output layer contains several nodes equal to the number of classes to be predicted.[51] Between these layers, there are weighted links,[9 50 51] the hidden layer receives a sum of the multiplication of the input variables with associated weights values plus the bias.[50 51] This value is entered into an activation function, such as a logistic or sigmoid function, to decide the class prediction. Outputs of the network are interpreted as class probabilities and sum to one.[51] We used nnet package to construct the ANN model.

RF is an extension of classification and regression trees.[9 10 51 52] RF performance is better compared with bagged trees because it decorrelates the trees,[53] hence improves accuracy.[52] Several forests of decision trees are grown using a random bootstrapped training sample. Also, instead of using all the variables/features in each tree, a random sample of variables are selected and tested at each split in each tree.[10 51 52] The prediction is made for unobserved data by taking a majority vote of the individual trees.[51 52] We used *randomForest* package to construct the RF model. NB is an effective classifier[50] due to its simplicity, exhibiting a surprisingly competitive predictive accuracy.[54] NB uses probability theory to find the most possible sample class in a classification problem. NB has two assumptions: (1) each attribute is conditionally independent of the other attributes given the class and (2) all the attributes have an impact on the class.[51 54] We used naivebayes package to construct the NB model.

Lreg is a standard multivariate classification method. It arises from the desire to model the posterior probabilities via linear functions in covariates, such that besides predicting class labels, it provides a probabilistic interpretation of this labeling.[53 55 56] Lreg uses a sigmoid function instead of a linear function to map predictions to probabilities between 0 and 1.[53] We used glm method to construct the Lreg model. Bagging, or bootstrap aggregation and boosting are general techniques for improving prediction rules and accuracy of the resulting predictions, by reducing the associated variance of prediction.[53 57] Bagging divides the available data into many bootstrap samples and train a separate model for each bootstrap, and then make a final prediction by averaging and voting for regression and classification, respectively.[57] Boosting, on the other hand, is a committee-based approach that uses a weighted average of prediction from various samples. The incorrectly predicted cases from a given step are given a higher weight during the next step. Thus, it is an iterative procedure, incorporating weights, as opposed to simple averaging of predictions.[57] We used *treebag* method and gbm package to construct the bagging and boosting models, respectively.

In the training set, parameter tuning and cross-validation aim to find a balance between building a model that can classify the training data effectively without overfitting to

the random fluctuations.[51] For each ML model, we used 10-fold cross-validation as a resampling method, where the training set is divided equally into 10 parts (folds). Therefore, every nine folds are used together for training the model and the remaining onefold for testing. This training-testing process is repeated ten times. We performed feature selection using the RF algorithm. After selecting the most important features, we retained in the dataset and used them for analysis in both the training and testing data for all models. We used the Synthetic Minority Over-sampling Technique (SMOTE) method[58 59] to address the class imbalance in the outcome (ie, the low proportion of perinatal deaths), by specifying the additional sampling to be 'smote' on train control parameter specifications. SMOTE is a method that produces artificial minority samples by interpolating between existing minority samples and their nearest minority neighbors.[58 59]

Using the testing set (30% of the remaining randomly selected sample), we computed the predictive performance of the six models (including Lreg model) from the training set using the area under the receiver-operating-characteristics curve (AUC ROC). We used the *ROCR* package for plotting ROC curves, obtaining the AUC values and comparison of models using AUC values. We also used measures from the confusion matrix results (ie, accuracy, sensitivity, specificity, positive and negative predictive values (NPV)), and the net benefit through decision curve analysis[60 61]—which quantifies whether a machine provides a relevant improvement in the prediction. We used epiR package to obtain confidence intervals for the performance measures and DCA package (http://www.decisioncurveanalysis.org) for decision curve analysis. We further used ggplot2 package to plot the decision curves. A good model will have a higher net benefit.[60] We used Delong's test to compare the ROC between models, where, a p<0.05 was considered statistically significant. The variable importance is a scaled measure with a maximum value of 100.[17] The R code used for this analysis is shown in online supplemental material 1.

## Patient and public involvement
There was no patient and public involvement.

## RESULTS
### Characteristics of study participants
The characteristics of the participants are shown in table 1. A total of 55 003 total deliveries were recorded at the KCMC medical birth registry from 2000 to 2015. Of these, we excluded 3316 (6%) multiple gestations (to avoid over-representation of high-risk pregnancies),[31] 49 (0.1%) records missing maternal identification numbers (hence could not be linked to child records), 791 (1.4%) records with a mismatch between the date of birth and unknown sequence (ie, singleton vs multiple births). We further excluded a total of 8528 (15.5%) observations with missing values in both the outcome (perinatal

status) and covariates. We, therefore, analysed data for a total of 42 319 singleton deliveries with complete records (figure 1).

The overall proportion of perinatal death among 42 319 singleton deliveries in this study was 3.7%. The proportion of perinatal deaths among mothers aged 20–34, 35–39 and 40+ years was 3.5%, 4.7% and 5.8%, respectively. Mothers with no education (5.6%) and those with primary education level (4.4%), who resided in rural areas (4.8%), had less than four ANC visits (5.9%), and those referred for delivery (7.8%) had a higher proportion of perinatal death. Among fathers, a higher proportion of perinatal death is among those aged 35+years (4.1%), with no (9.6%) or with primary education level (4.6%) as well as those who were unemployed (5.7%), table 1.

Furthermore, the most common obstetric care and complications in this birth cohort included induction of labour (22.7%), malaria (13.2%), preterm birth (10.8%) and LBW (10.2%). About 4% of mothers in this cohort experienced pre-eclampsia/eclampsia during pregnancy. Less than half of all children were females. The proportion of perinatal death among women who experienced induction of labour, with malaria, delivered preterm, delivered LBW baby and experienced pre-eclampsia/eclampsia during pregnancy was 4.8%, 3.8%, 14.2%, 17.6% and 12.5%, respectively. The proportion of perinatal death is almost similar among males (3.8%) compared with females (3.6%) children in this cohort (table 1).

The trends in the proportion of perinatal deaths that occurred at KCMC between the years 2000–2015 are shown in figure 2. Overall, the proportion of perinatal deaths has slightly declined over the years by 6% (95% CI, 0.3% to 12.3%), though this decline was not statistically significant (p=0.06).

### Variable importance
We used the RF algorithm for feature/variable selection. This model selected a total of 20 important predictors (figure 3) based on its threshold measure of importance out of the 32 variables. We used these 20 variables in all the subsequent analysis for all models in both training and testing sets.

### Predicting perinatal deaths
The discriminatory abilities of all models for the prediction of perinatal death are in figure 4A and table 2. There were no significant differences (p>0.05) in the AUC the ROC curve between Lreg with RF, ANN, boosting and NB. However, bagging had significantly lower predictive performance (AUC 0.76, 95% CI 0.74 to 0.79, p=0.006) compared with the Lreg model (AUC 0.78, 95% CI 0.76 to 0.81). Furthermore, the ANN model (sensitivity 0.60, 95% CI 0.55 to 0.64) and NB model (Sensitivity 0.57, 95% CI 0.52 to 0.62) had slightly higher sensitivity compared with Lreg (sensitivity 0.56, 95% CI 0.51 to 0.60) while boosting (Specificity 0.89 95% CI 0.88 to 0.89) and RF

**Table 1** Characteristics of study participants (n=42 319)

| Characteristics | Total | Perinatal death | P value* | Characteristics | Total | Perinatal death | P value* |
|---|---|---|---|---|---|---|---|
| **Maternal** | n (%) | n (%) | | **Obstetrics** | n (%) | n (%) | |
| **Age (years)** | | | <0.001 | **Gestational diabetes** | | | 0.89 |
| 15–19 | 3470 (8.2) | 99 (2.9) | | No | 42 288 (99.9) | 1560 (3.7) | |
| 20–34 | 32 675 (77.2) | 1158 (3.5) | | Yes | 31 (0.1) | 1 (3.2) | |
| 35–39 | 4984 (11.8) | 235 (4.7) | | **Diabetes** | | | 0.002 |
| 40+ | 1190 (2.8) | 69 (5.8) | | No | 42 240 (99.8) | 1553 (3.7) | |
| **Education level** | | | <0.001 | Yes | 79 (0.2) | 8 (10.1) | |
| None | 567 (1.3) | 32 (5.6) | | **Hypertension** | | | <0.001 |
| Primary | 23 010 (54.4) | 1019 (4.4) | | No | 42 241 (99.8) | 1550 (3.7) | |
| Secondary | 5275 (12.5) | 159 (3.0) | | Yes | 78 (0.2) | 11 (14.1) | |
| Higher | 13 467 (31.8) | 351 (2.6) | | **Bleeding** | | | <0.001 |
| **Occupation** | | | 0.37 | No | 41 897 (99.0) | 1528 (3.6) | |
| Unemployed | 9316 (22.0) | 365 (3.9) | | Yes | 422 (1.0) | 33 (7.8) | |
| Employed | 30 061 (71.0) | 1085 (3.6) | | **Anaemia** | | | 0.004 |
| Others | 2942 (7.0) | 111 (3.8) | | No | 41 661 (98.4) | 1523 (3.7) | |
| **Marital status** | | | 0.89 | Yes | 658 (1.6) | 38 (5.8) | |
| Single | 4954 (11.7) | 186 (3.8) | | **Malaria** | | | 0.79 |
| Married | 37 300 (88.1) | 1372 (3.7) | | No | 36 746 (86.8) | 1352 (3.7) | |
| Widowed/divorced | 65 (0.2) | 3 (4.6) | | Yes | 5573 (13.2) | 209 (3.8) | |
| **Area of residence** | | | <0.001 | **Sepsis/infections** | | | 0.43 |
| Urban | 25 056 (59.2) | 725 (2.9) | | No | 41 588 (98.3) | 1538 (3.7) | |
| Rural | 17 263 (40.8) | 836 (4.8) | | Yes | 731 (1.7) | 23 (3.1) | |
| **Alcohol consumption during pregnancy** | | | 0.001 | **Complications** | | | |
| No | 30 759 (72.7) | 1191 (3.9) | | **Pre-eclampsia/ eclampsia** | | | <0.001 |
| Yes | 11 560 (27.3) | 370 (3.2) | | No | 40 668 (96.1) | 1355 (3.3) | |
| **Smoking during pregnancy** | | | 0.97 | Yes | 1651 (3.9) | 206 (12.5) | |
| Yes | 53 (0.1) | 2 (3.8) | | **Induction of labour** | | | <0.001 |
| No | 42 266 (99.9) | 1559 (3.7) | | No | 32 732 (77.3) | 1105 (3.4) | |
| **No of ANC visits** | | | <0.001 | Yes | 9587 (22.7) | 456 (4.8) | |
| ≥4 | 28 742 (67.9) | 760 (2.6) | | **PROM** | | | 0.006 |
| <4 | 13 577 (32.1) | 801 (5.9) | | No | 41 416 (97.9) | 1543 (3.7) | |
| **Referred for delivery** | | | <0.001 | Yes | 903 (2.1) | 18 (2.0) | |
| No | 32 762 (77.4) | 819 (2.5) | | **PPH** | | | <0.001 |
| Yes | 9557 (22.6) | 742 (7.8) | | No | 42 091 (99.5) | 1516 (3.6) | |
| **Paternal characteristics** | | | | Yes | 228 (0.5) | 45 (19.7) | |
| **Age (years)** | | | 0.001 | **3–4 degree tear** | | | 0.49 |
| <25 | 3938 (9.3) | 122 (3.1) | | No | 42 305 (99.9) | 1560 (3.7) | |
| 25–29 | 10 593 (25.0) | 346 (3.3) | | Yes | 14 (0.1) | 1 (7.1) | |

Continued

**Table 1** Continued

| Characteristics | Total | Perinatal death | P value* | Characteristics | Total | Perinatal death | P value* |
|---|---|---|---|---|---|---|---|
| **Maternal** | n (%) | n (%) | | **Obstetrics** | n (%) | n (%) | |
| 30–34 | 12 303 (29.1) | 457 (3.7) | | **Abruption placenta** | | | <0.001 |
| 35+ | 15 485 (36.6) | 636 (4.1) | | No | 42 193 (99.7) | 1490 (3.5) | |
| **Education level** | | | <0.001 | Yes | 126 (0.3) | 71 (56.3) | |
| None | 281 (0.7) | 27 (9.6) | | **Placenta previa** | | | 0.04 |
| Primary | 18 987 (44.9) | 868 (4.6) | | No | 42 245 (99.8) | 1555 (3.7) | |
| Secondary | 4565 (10.8) | 154 (3.4) | | Yes | 74 (0.2) | 6 (8.1) | |
| Higher | 18 486 (43.7) | 512 (2.8) | | **Presentation** | | | <0.001 |
| **Occupation** | | | <0.001 | Cephalic | 41 833 (98.9) | 1459 (3.5) | |
| Unemployed | 5710 (13.5) | 323 (5.7) | | Breach/ Transverse | 486 (1.1) | 102 (21.0) | |
| Employed | 36 102 (85.3) | 1218 (3.4) | | **Gestational age at birth** | | | <0.001 |
| Others | 507 (1.2) | 20 (3.9) | | Term birth (≥37 weeks) | 37 764 (89.2) | 914 (2.4) | |
| | | | | Preterm birth (<37 weeks) | 4555 (10.8) | 647 (14.2) | |
| | | | | **Birth weight** | | | <0.001 |
| | | | | Normal birth weight | 37 991 (89.8) | 801 (2.1) | |
| | | | | Low birth weight | 4328 (10.2) | 760 (17.6) | |
| | | | | **Child's sex** | | | 0.42 |
| | | | | Female | 20 430 (48.3) | 738 (3.6) | |
| **Total** | 42 319 | 1561 (3.7%) | | Male | 21 889 (51.7) | 823 (3.8) | |

*P value based on the $\chi^2$ test.
ANC, antenatal care; PPH, postpartum haemorrhage; PROM, premature rupture of the membranes.

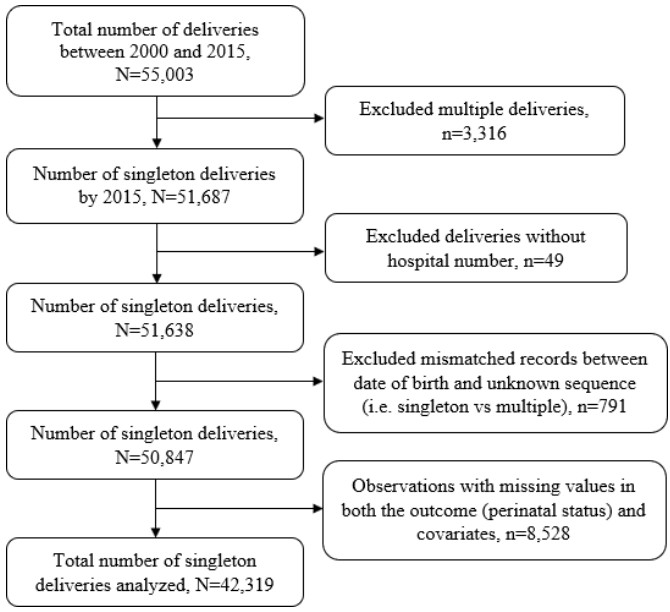

**Figure 1** Schematic diagram showing the number of singleton deliveries analysed, KCMC medical birth registry data, 2000–2015. KCMC, Kilimanjaro Christian Medical Centre.

(Specificity 0.88, 95% CI 0.88 to 0.89) had slightly higher specificity compared with Lreg (specificity 0.87, 95% CI 0.86 to 0.88). Due to the low prevalence of perinatal

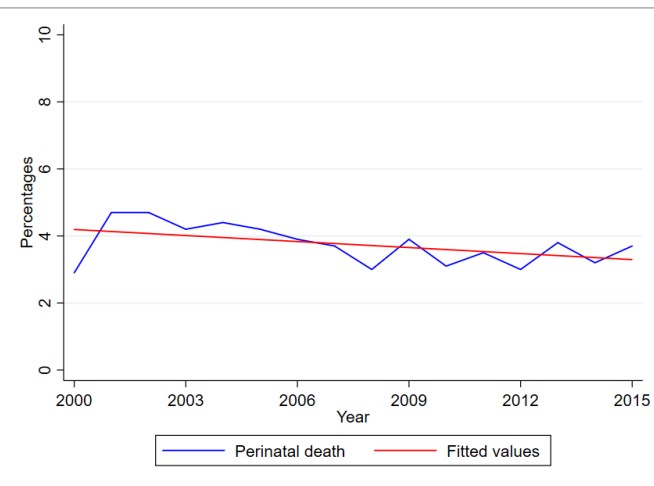

**Figure 2** Trends of perinatal death, KCMC medical birth registry data, 2000–2015. KCMC, Kilimanjaro Christian Medical Centre.

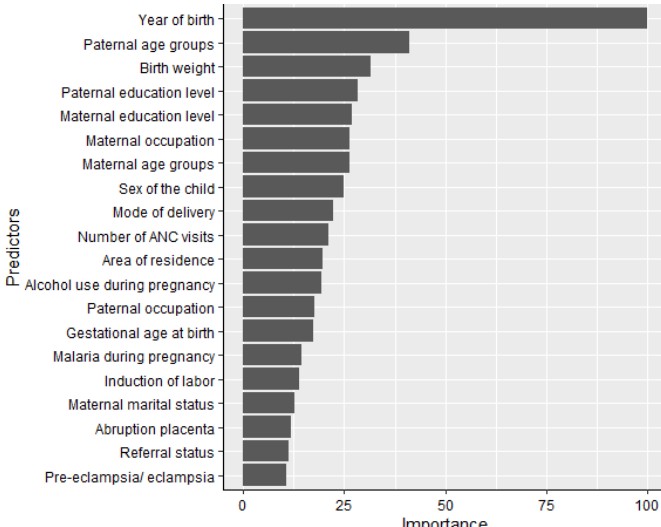

**Figure 3** Variable importance of predictors for perinatal death in the random forest model scaled to have a maximum value of 100. ANC, antenatal care.

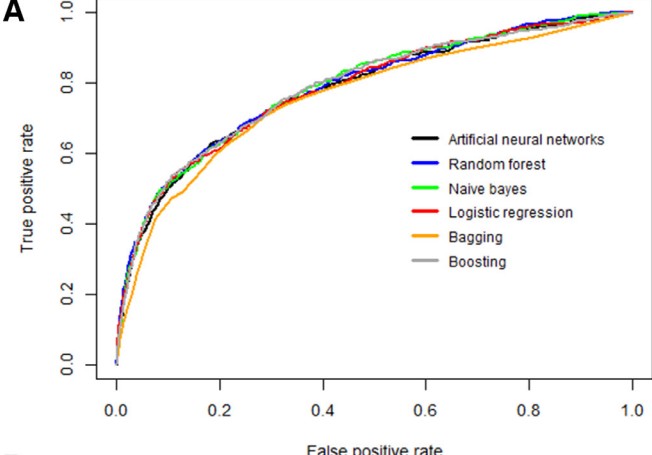

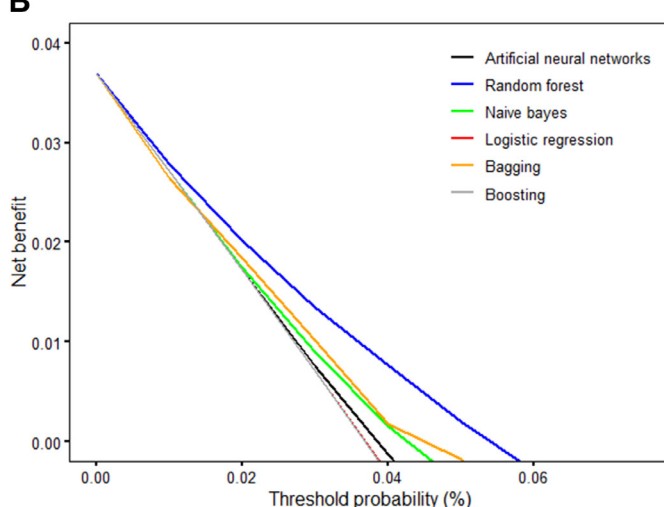

**Figure 4** Prediction ability of perinatal deaths comparing different machine learning models in the test set: (A) Receiver operating characteristics curves. The corresponding values of the area under the receiver operating characteristics curve for each model are in table 2. (B) Decision curve analysis. The net benefit of the machine learning models (except for boosting) is larger over a range of threshold probability values compared with that of the logistic regression model.

deaths (3.7%), all models had high (NPV 0.98, 95% CI 0.98 to 0.98).

With regard to the number of actual and predicted outcomes (table 3), all models correctly predicted perinatal deaths by more than half of 468 deaths in the testing set. The numbers of correct classification were higher in the ANN 280 (59.8%) and NB 267 (57.1%), followed by the Lreg model 261 (55.8%) and bagging 260 (55.6%). The decision curve analysis (figure 4B) demonstrated that the net benefit of the RF model surpassed that of other ML models, including Lreg for all threshold values, indicating that the RF model is more superior in predicting the risk of perinatal deaths in this cohort. The accuracy of the RF model was 0.87, 95% CI (0.86 to 0.87), compared with 0.87, 95% CI (0.87 to 0.88) for boosting and 0.86, 95% CI (0.85 to 0.86) for the Lreg model (table 2). Furthermore, other ML models (except for boosting) demonstrated high net benefit over a range of threshold probability values relative to that of the Lreg model. Also, the RF model had a superior net benefit over all models (figure 4B).

## DISCUSSION

In this study, the perinatal death was predicted using five ML models (ANN, RFs, NB, bagging and boosting). There were no differences in the predictive performance between ML models except for bagging, which had a lower predictive performance. The ANN and NB had higher sensitivity compared with the Lreg, and other ML models. Specificity for all models was high, mainly due to the low prevalence of perinatal deaths in this cohort. Additionally, results from the decision curve analysis revealed that the ML models (except for boosting) had a higher net benefit over a range of threshold probability values compared with the Lreg model, indicating high

accuracy. The RF model demonstrated a superior net benefit over other models.

In the present study, maternal characteristics before and during pregnancy, pregnancy history, and paternal characteristics identified pregnancies at high risk of experiencing adverse perinatal outcomes that might need close clinical follow-ups. It is worth noting here that paternal age and education level were highly predictive of perinatal death more than the known pregnancy-related conditions or complications such as prematurity. Previous literature shows that paternal characteristics, particularly advanced paternal age, increase the risk of adverse perinatal outcomes, such as low birth weight, prematurity, small for gestational age and low Apgar scores,[62–64] despite conflicting evidence from other studies.[65] Furthermore, studies using data from the KCMC Medical Birth Registry (same data source to the current study) focused on modelling the association between maternal

**Table 2** Prediction performance of the reference and machine learning models in the test set

| Model | Logistic regression | Artificial neural network | Random forests | Naïve bayes | Bagging | Boosting |
|---|---|---|---|---|---|---|
| ACC | 0.86 (0.85 to 0.86) | 0.83 (0.82 to 0.83) | 0.87 (0.86 to 0.87) | 0.84 (0.83 to 0.85) | 0.82 (0.81 to 0.83) | 0.87 (0.87 to 0.88) |
| AUC | 0.78 (0.76 to 0.81) | 0.78 (0.76 to 0.80) | 0.79 (0.76 to 0.81) | 0.79 (0.76 to 0.81) | 0.76 (0.74 to 0.79) | 0.79 (0.76 to 0.81) |
| P value* | Reference | 0.59 | 0.37 | 0.65 | 0.006 | 0.20 |
| Sensitivity | 0.56 (0.51 to 0.60) | 0.60 (0.55 to 0.64) | 0.54 (0.49 to 0.58) | 0.57 (0.52 to 0.62) | 0.55 (0.50 to 0.59) | 0.54 (0.49 to 0.58) |
| Specificity | 0.87 (0.86 to 0.88) | 0.84 (0.83 to 0.84) | 0.88 (0.88 to 0.89) | 0.85 (0.84 to 0.86) | 0.83 (0.82 to 0.84) | 0.89 (0.88 to 0.89) |
| PPV | 0.14 (0.12 to 0.16) | 0.12 (0.11 to 0.14) | 0.15 (0.13 to 0.17) | 0.13 (0.11 to 0.14) | 0.11 (0.10 to 0.12) | 0.15 (0.14 to 0.17) |
| NPV | 0.98 (0.98 to 0.98) | 0.98 (0.98 to 0.98) | 0.98 (0.98 to 0.98) | 0.98 (0.98 to 0.98) | 0.98 (0.98 to 0.98) | 0.98 (0.98 to 0.98) |

*We calculated p values to compare the AUC the receiver operating characteristics curve of logistic with each machine learning model.
ACC, accuracy; AUC, area under the curve; NPV, negative predictive value; PPV, positive predictive value.

**Table 3** The number of actual and predicted outcomes of prediction models in the test set

| Prediction model | Classification | Perinatal status | |
|---|---|---|---|
| | | Alive | Died |
| | Actual number of events | 12 227 | 468 |
| Logistic regression | Correctly predicted outcome | 10 627 | 261 |
| | Incorrectly predicted outcome | 1600 | 207 |
| Artificial neural network | Correctly predicted outcome | 10 225 | 280 |
| | Incorrectly predicted outcome | 2002 | 188 |
| Random Fforests | Correctly predicted outcome | 10 774 | 251 |
| | Incorrectly predicted outcome | 1453 | 217 |
| Naïve bayes | Correctly predicted outcome | 10 386 | 267 |
| | Incorrectly predicted outcome | 1841 | 201 |
| Bagging | Correctly predicted outcome | 10 175 | 260 |
| | Incorrectly predicted outcome | 2052 | 208 |
| Boosting | Correctly predicted outcome | 10 852 | 252 |
| | Incorrectly predicted outcome | 1375 | 216 |

and pregnancy-related characteristics and complications during pregnancy and childbirth with the risk of adverse perinatal outcomes[26–31] but ignored paternal characteristics. Despite challenges in male involvement in pregnancy and childbirth in Tanzania,[66 67] their participation is critical to improving maternal and child health outcomes.

On top of clinicians' judgement, previous investigators applied standard regression models in prediction of risk for adverse perinatal outcomes, particularly perinatal death.[29 39–45 68–73] We found no differences in the predictive performance of the ML models, except for bagging, which had lower predictive capacity. The sensitivity of the ML models was also almost comparable to that of Lreg, which indicates that both models correctly classified perinatal deaths. Our finding is consistent with a recent systematic review that showed no performance benefit of ML models over Lreg for the prediction of clinical outcomes.[19] The possible explanation for lack of differences in the performance between the compared models could be attributed to the low proportion of outcome and exposures in this study, as well as data quality and recording challenges inherent in registry-based studies.

In contrast, some previous investigators have demonstrated that ML models offer better predictions of clinical

or adverse pregnancy outcomes compared with classical regression models.[13–16] The application of ML models may improve the classification of adverse events occurring during the perinatal period and, therefore, assist in triaging and provision of close clinical follow-up for women at high risk. Other studies also provide evidence of improved prediction of under-5 and neonatal mortality[12–15] using ML models. The utility of these models may, therefore, improve the prediction of adverse pregnancy outcomes as opposed to standard regression models.

In this study, the decision curve analysis that accounts for the impact of false-negative and false-positive misclassification errors showed superior predictive performance of the ML approaches over the Lreg model. This demonstrates a higher net benefit for the prediction of perinatal deaths. The higher net benefit in the prediction ability of the ML approaches has also been documented elsewhere.[10 17] This is because ML approaches can incorporate the high order nonlinear interactions between predictors, which cannot be addressed by traditional modelling approaches, including the Lreg model. Furthermore, the use of cross-validation is also known to reduce potential overfitting in ML models. It is important to note that ML approaches are, to a large extent, nonparametric as opposed to the Lreg model that relies on strong distributional assumptions.

The strength of this study is that it is the first to apply modern ML approaches to predict perinatal deaths, particularly in Tanzania and to a large extent sub-Saharan Africa, compared with the classical Lreg model. Our study demonstrated that ML models might be used to improve the prediction of perinatal deaths and triage of women at risk. We also used the SMOTE balancing technique to avoid the bias of the model toward skewed data (reduce overfitting), hence improving the prediction accuracy of the ML algorithm.[14 58 59] However, SMOTE is not very effective for high dimensional data.[74 75] Our study also had some limitations that are worth considering when interpreting the results. First, we excluded observations with missing values in both the outcome and exposures from the analysis, a problem inherent in cohort studies, including birth registries, which may lead to underestimation of the proportion of perinatal death. Two excluded variables (maternal BMI and HIV status) have been associated with perinatal and under-5 deaths[4 6 38]; hence their exclusion might increase the risk of residual confounding bias. The effect of exclusion of these two variables and missing values to predict perinatal deaths remains unquantified.

Second, selection bias/or referral bias is a common problem in hospital-based studies, which affects the generalisation of findings to the general population. This might also be the case in the present study. However, our findings might reflect a similar setting in Tanzania and probably in other sub-Saharan African countries. Third, the KCMC Medical Birth Registry cohort only captures perinatal deaths occurring in the health facility (KCMC hospital), which may underestimate the observed perinatal deaths in the wider population. Currently, the hospital has no mechanisms to follow-up the birth outcomes from deliveries that occur at home and post-discharge outcomes of the babies after mothers are discharged from the hospital within the first week, especially within the KCMC hospital catchment area. Future extensions include ways of handling missing values before applying the ML algorithms to predict perinatal death and other adverse pregnancy outcomes.

## CONCLUSION

The ML models (except for bagging) performed equally with the Lreg model to predict perinatal deaths using maternal, paternal and obstetric factors in this cohort. The ML models, however, have a higher net benefit, demonstrating superiority in the prediction of perinatal death. Furthermore, the RF model also demonstrated superior performance over other ML models. These models are a useful and alternative strategy over the standard Lreg model to predict perinatal deaths, considering the richness of the medical birth registries. Moreover, the ML models are capable of handling many predictors at the same time, which is crucial in capturing multiple risk factors for adverse perinatal outcomes such as perinatal deaths. The application of ML models may, therefore, increase the prediction ability of adverse perinatal outcomes and thereby helping in triage women most at risk.

**Acknowledgements** We would like to acknowledge the midwives who participated in data collection and all women and children whose information enabled the availability of data used in this study. The authors also thank the staff at the Birth Registry for capturing these data in the electronic system. We also appreciate the Centre for International Health at the University of Bergen in Norway and the Department of Obstetrics and Gynecology of the KCMC hospital in Tanzania for establishing the KCMC medical birth registry, which facilitated the availability of data to conduct this study.

**Contributors** All authors made a substantial contribution to this study. IBM, MJM, JO and HGM designed the study. IBM and MM analysed the data. MJM, JO and HGM commented on the manuscript. IBM drafted the manuscript and had primary responsibility for the final content. All authors reviewed the drafts of this manuscript and approved the final version for submission.

**Funding** This work was funded by GSK Africa Non-Communicable Disease Open Lab through the DELTAS Africa Sub-Saharan African Consortium for Advanced Biostatistics (SSACAB) Grant No. 107754/Z/15/Z- training programme. The views expressed in this publication are those of the author(s) and not necessarily those of GSK.

**Disclaimer** The views expressed in this publication are those of the author(s) and not necessarily those of AAS, NEPAD Agency, Wellcome Trust or the UK government. The Norwegian Council for Higher Education's Programme for Development Research (NUFU) funded the establishment of the birth registry at the Kilimanjaro Christian Medical Centre (KCMC). The funders had no role in study design, data collection, and analysis, decision to publish, or preparation of the manuscript.

**Competing interests** None declared.

**Patient consent for publication** Not required.

**Ethics approval** This study was approved by the Kilimanjaro Christian Medical College Research Ethics and Review Committee (KCMU-CRERC) with approval number 2424. Before data collection, mothers provided oral informed consent after receiving information about the purpose of the study, questionnaire, and its contents as well as the intention to gather new knowledge, which will, in turn,

benefit mothers and children in the future. Mothers were informed that participation was voluntary and had no implications on the care they would receive. Following consent, mothers were free to refuse to reply to single questions. For privacy and confidentiality, unique identification numbers were used to identify and then link mothers with child records. There was no person-identifiable information in any electronic database, and instead, unique identification numbers were used. Necessary measures were taken by midwives to ensure privacy during the interview process.

**Provenance and peer review**  Not commissioned; externally peer reviewed.

**Data availability statement**  Data are available on reasonable request. The KCMC medical birth registry data contains potentially identifying and sensitive patient information. This has also been stipulated by the local Institutional Review Board of KCMC hospital and the National Ethics Committee in Norway when establishing this birth registry. Permission to use the data in this study was made through the Kilimanjaro Christian Medical University College Research and Ethics Review Committee, and received an approval number 2424. The authors do not have the legal right to share the data publicly. All data requests can be sent to the Executive Director of the KCMC referral hospital, P. O. Box 3010, Moshi, Tanzania, Email: kcmcadmin@kcmc.ac.tz or through the corresponding author.

**ORCID iDs**
Innocent B Mboya http://orcid.org/0000-0001-9861-5879
Michael J Mahande http://orcid.org/0000-0002-7750-7657

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
