## [Reviewer comments · BMJ Open]

ARTICLE DETAILS

TITLE (PROVISIONAL)	Prediction of perinatal death using machine learning models: A birth registry-based cohort study in northern Tanzania
AUTHORS	Mboya, Innocent; Mahande, Michael Johnson; Mohammed, Mohanad; Obure, Joseph; Mwambi, Henry

VERSION 1 – REVIEW

REVIEWER	David Edwards King's College London United Kingdom
REVIEW RETURNED	19-May-2020

GENERAL COMMENTS	I enjoyed reading this paper and I think it is a useful study. The fundamental finding is that machine learning is not a magic trick that will extract new clinical information from this large and valuable dataset- which in itself is worth knowing. I can see good reason for trying to move away from logistic regression for data with strong co-dependencies, but it would seem that these methods add little. I feel there are some insight which the analysis provides which could be drawn out further by the researchers. I dont have any comments to make on the machine learning methods, which seem appropriate, and 10-fold cross validation seems the right way to build the models. I did wonder if one of the network clustering approaches might be better, but that is a different paper and the authors may be doing that elsewhere. First, it would appear that the data collected may not contain the information needed to make the predictions they what to make. My guess is that there are important data that have not been recorded. HIV and BMI are missing and excluded, and this would suggest that an important take home message is that better recording of these variables could be a priority, but in addition I would suspect that attention to the accurate measurement of maternal blood pressure - the rate of pre-eclampsia seems low. It would be an important contribution to women's health if better blood pressure monitoring was aided as a result of this analysis. I also note that paternal age seems highly predictive and this hints at social factors that I have not understanding of. And the relatively small contribution from prematurity as a predictor also seems at odds with experience elsewhere and I wonder if this is again because of social factors and the exclusion of out of hospital deaths. I think the authors could usefully comment in more detail on these issues.
---

	Second, just because there is no significant trend in the death rate by time does not mean this is not an important confound and I wonder if the authors though of regressing this out of their dataset? It is striking that is seems the strongest predictor in the Random Forest analysis.
--	--

REVIEWER	Kamran Afzali UdeM, Canada
REVIEW RETURNED	16-Jun-2020

GENERAL COMMENTS	This manuscript applied a machine-learning framework to predict perinatal death in a birth registry-based cohort study in northern Tanzania. The questions asked are relevant, however, I have a number of questions/suggestions for the authors that will hopefully improve the quality of the manuscript.  1. My first major concern is the potential lack of generalizability of the results presented here. I totally understand that the external sample generalizability can be irrelevant in the context of this research however I wonder if the was model trained on the data from 2005 to 2010, would it perform similarly on the data from 2010 to 2015. 2. The authors did not discuss or implement a nested cross validation procedure with grid search (for hyper parameters) to highlight potential robustness of their results. A grid search might improve the performance. 3. My second major concern is the lack of discussion or consideration around alternative models (e.g. SVC, Xgboost) for the same kind of modelling. What is the rationale behind the selected models? 4. It is unclear why the authors opted to ignore missing value imputation and just indicate that "Future extensions include ways of handling missing values before applying the machine learning algorithms.". Implementing an imputation strategy can highlight the robustness of the results. 5. Please explain the choice of hyper-parameters in your model (e.g. number of trees). 6. I commend the authors for their use of a robust SMOTE algorithm to address the class imbalance. However, as this method might increase the bias through over-fitting please indicate the potential downside of this methodology in the limitation section. 7. It is unclear to me why the presented variable importance analysis corresponds to the results of only one algorithm? Do feature importance analysis of other algorithms indicate consistent results? I strongly recommend the authors to explain the exact details of feature importance analysis for each algorithm. 8. Including the code (including preprocessing, predictive modelling, missing data imputation, cross validation, and feature importance analysis) in the supplementary materials could substantially increase the clarity and reproducibility of this study.
--

VERSION 1 – AUTHOR RESPONSE

Reviewer #1:

Reviewer Name: David Edwards

Institution and Country: King's College London, United Kingdom

Please state any competing interests or state 'None declared': none

1. I enjoyed reading this paper and I think it is a useful study. The fundamental finding is that machine learning is not a magic trick that will extract new clinical information from this large and valuable dataset- which in itself is worth knowing. I can see good reason for trying to move away from logistic regression for data with strong co-dependencies, but it would seem that these methods add little.

Response: We thank the reviewer for this important observation and comment. The findings from this study indicate that there are no significant differences in the predictive performance of the machine learning methods compared to logistic regression, except for bugging, which had low predictive performance. However, the machine learning methods had a higher net benefit for correct classification of perinatal deaths compared to the logistic regression model. Despite the conflicting evidence from other literature of the predictive performance of the machines, the application of machine learning models depends on the research question and the context. Yet, these methods are shown to improve the predictive performance of different classification problems, especially in high-dimension data.

2. I feel there are some insight which the analysis provides which could be drawn out further by the researchers.

Response: We thank the reviewer for this comment and suggestion. We hope the revisions in the manuscript can emphasize further the relevance and implications of findings from this study.

3. I don't have any comments to make on the machine learning methods, which seem appropriate, and 10-fold cross validation seems the right way to build the models. I did wonder if one of the network clustering approaches might be better, but that is a different paper and the authors may be doing that elsewhere.

Response: We thank the reviewer for the thought-provoking comments and insights. In this dataset, we applied the supervised machine learning algorithms as opposed to clustering approaches where the classes or labels are not known. We modelled perinatal death status containing class labels with 32 labelled features/ variables, except for survey year. However, as rightfully indicated by the reviewer, we will consider exploring the unsupervised or semi-supervised algorithms in future work to assess whether we will obtain different conclusions would from such analyses.

4. First, it would appear that the data collected may not contain the information needed to make the predictions they what to make. My guess is that there are important data that have not been recorded. HIV and BMI are missing and excluded, and this would suggest that an important take home message is that better recording of these variables could be a priority, but in addition I would suspect that attention to the accurate measurement of maternal blood pressure - the rate of pre-eclampsia seems low. It would be an important contribution to women's health if better blood pressure monitoring was aided as a result of this analysis.

Response: We acknowledge the reviewer's comment. Indeed, the Medical Birth Registry at KCMC referral hospital was established in the year 2000. During this time, there was no internationally approved classification of deaths occurring during the perinatal period, which came 16 years later. It is therefore possible that, the key predictors such as those related to the health system are not collected in the medical registry (missed). Because of that and many other factors, the registry requires immediate updates to reflect the new guidelines also based on evidence from the current literature. Efforts are underway to resolve data quality challenges in the birth registry necessary to increase accuracy. If this work is accepted for publication, these findings will contribute to information required for strategic decision making. Beyond that, the measurement errors related to different variables in the dataset is something beyond the control of this study, given also we utilized secondary data.

5. I also note that paternal age seems highly predictive and this hints at social factors that I have not understanding of, and the relatively small contribution from prematurity as a predictor also seems at odds with experience elsewhere and I wonder if this is again because of social factors and the exclusion of out of hospital deaths. I think the authors could usefully comment in more detail on these issues.

Response: We acknowledge the reviewer comment. We have revised the discussion section to include information about the need to consider paternal characteristics during prenatal, labour and postnatal care to improving maternal and child health outcomes. The revised information is in the second paragraph of the discussion section.

6. Second, just because there is no significant trend in the death rate by time does not mean this is not an important confound and I wonder if the authors thought of regressing this out of their dataset? It is striking that it seems the strongest predictor in the Random Forest analysis.

Response: We thank the reviewer for this comment. We would, however, like to indicate that, the point estimate, confidence intervals and the level of significance reported below Table 1 (related to Figure 2) on page 13 were based on results of regressing perinatal status with year as a covariate of interest. We modelled year on a continuous scale in the machine learning models. The strongest prediction of perinatal deaths in the Random Forest analysis would be in line with the declining trends based on the fact that significant progress was made between 2000-2015 (the MDG era) in improving child survival.

Reviewer #2:

Reviewer Name: Kamran Afzali

Institution and Country: UdeM, Canada

Please state any competing interests or state 'None declared': 'None declared'

This manuscript applied a machine-learning framework to predict perinatal death in a birth registry-based cohort study in northern Tanzania. The questions asked are relevant, however, I have a number of questions/suggestions for the authors that will hopefully improve the quality of the manuscript.

1. My first major concern is the potential lack of generalizability of the results presented here. I totally understand that the external sample generalizability can be irrelevant in the context of this research, however I wonder if there was a model trained on the data from 2005 to 2010, would it perform similarly on the data from 2010 to 2015.

Response: We thank the reviewer for this important comment and suggestion. The model was trained for data from 2000 to 2015. We didn't perform a sub-analysis to train the model for different groups of years. Because this problem has never been explored in this birth registry using machine learning models, we aimed to apply these methods to predict perinatal death and show whether findings from this study will be in line with those utilizing the same data source using classical logistic regression models. We have revised the discussion section to briefly contrast the results from this study, and those that also used data from the KCMC medical birth registry. The idea of splitting the data in two parts; 2005-2010 and 2011-2015 is something we had not thought about. We will, however, explore this approach in further analysis.

2. The authors did not discuss or implement a nested cross validation procedure with grid search (for hyper parameters) to highlight potential robustness of their results. A grid search might improve the performance.

Response: We acknowledge the reviewer comment. However, in our analysis, we used the train function from caret package in R, which provides a grid search where the function itself can specify

the parameters of the models. The train function tries all the combinations of parameters values and selects the ones that give the best performance. The parameters were, therefore, not tuned manually. While the application of a nested cross-validation with grid search was beyond the scope of this work, we will consider the reviewer comment in a follow-up study.

3. My second major concern is the lack of discussion or consideration around alternative models (e.g. SVC, Xgboost) for the same kind of modelling. What is the rationale behind the selected models?

Response: Although there are several models used for analyzing such a problem, we used six different ML models to predict perinatal death, which has several merits (as also described in the statistical and computational analysis section). Furthermore, the performance indicators, particularly the AUC, showed no significant differences between the logistic regression model and the machine learning algorithms. However, the later (machine learning algorithms) had a higher net-benefit for correct classification of perinatal deaths. Despite that, the application of SVC and Xgboost among many other machine learning algorithms is an area we can further explore using the KCMC Medical Birth Registry data. In fact, in this study, two additional classifiers; SVM and LDA did not fit the problem of interest. Because of that, we saw no need to report anything regarding the application of the two methods.

4. It is unclear why the authors opted to ignore missing value imputation and just indicate that “Future extensions include ways of handling missing values before applying the machine learning algorithms.”. Implementing an imputation strategy can highlight the robustness of the results.

Response: We agree with the reviewer that implementing an imputation strategy can highlight the robustness of the results. However, we would like to emphasize that the primary focus of this paper is the application of machine learning algorithms to predict perinatal death. This is based on the background that, previous studies using the same data source have never applied such methods to predict adverse pregnancy outcomes such as perinatal death. We aim to show the added benefit of using these models as opposed to standard regression analysis like the logistic regression model, which has been extensively applied to such a problem. The findings from this study have shown that machine learning methods can improve the classification of the adverse pregnancy outcomes, particularly perinatal death.

5. Please explain the choice of hyper-parameters in your model (e.g. number of trees).

Response: As indicated earlier, the hyper-parameters were not tuned manually but tuned using the caret package, which tries several combinations of options to arrive at the model with the best performance. We have revised the methods section to make this clearer (see the revised section).

6. I commend the authors for their use of a robust SMOTE algorithm to address the class imbalance. However, as this method might increase the bias through over-fitting please indicate the potential downside of this methodology in the limitation section.

Response: We thank the reviewer for this observation. SMOTE algorithm aims to reduce over-fitting in low dimensional data. This technique is, however, not effective for high dimensional data. We have revised the discussion section to address this limitation.

7. It is unclear to me why the presented variable importance analysis corresponds to the results of only one algorithm? Do feature importance analysis of other algorithms indicate consistent results? I strongly recommend the authors to explain the exact details of feature importance analysis for each algorithm.

Response: We acknowledge the reviewer comment. Random forest algorithm was used for feature selection whereby the most important (20 features) were used in training and testing steps of the analysis. Same features were used for all the models. We have revised the statistical and computational analysis section of the Methods, in the last paragraph of page 8. We have also added this information in the results, variable importance section. The reason for using this algorithm for feature selection is because the literature shows that RF performs best for both classification and regression problems. Also, RF is robust to over-fitting, is able to handle highly non-linear data, among other advantages.

8. Including the code (including preprocessing, predictive modelling, missing data imputation, cross validation, and feature importance analysis) in the supplementary materials could substantially increase the clarity and reproducibility of this study.

Response: We acknowledge the reviewer comment. We have attached the code used in the analysis as the supplementary materials. Because we did not analyze the missing data, no such information is contained in the code.